# Efficacy of Conversion Surgery for Initially Unresectable Biliary Tract Cancer That Has Responded to Down-Staging Chemotherapy

**DOI:** 10.3390/cancers17050873

**Published:** 2025-03-03

**Authors:** Takashi Murakami, Ryusei Matsuyama, Yasuhiro Yabushita, Yuki Homma, Yu Sawada, Kentaro Miyake, Takafumi Kumamoto, Kazuhisa Takeda, Shin Maeda, Shoji Yamanaka, Itaru Endo

**Affiliations:** 1Department of Gastroenterological Surgery, Graduate School of Medicine, Yokohama City University, Yokohama 236-0004, Japan; gtrennsport3@gmail.com (T.M.); ryusei@yokohama-cu.ac.jp (R.M.);; 2Department of Gastroenterology, Graduate School of Medicine, Yokohama City University, Yokohama 236-0004, Japan; 3Department of Pathology, Yokohama City University Hospital, Yokohama 236-0004, Japan

**Keywords:** conversion surgery, down-staging chemotherapy, unresectable biliary tract cancer, down-staging chemotherapy, overall survival

## Abstract

We performed conversion surgery in patients with unresectable biliary tract cancer who responded to down-staging chemotherapy, and the 5-year survival rate was significantly higher in patients who underwent conversion surgery than in those who did not. The importance of this finding is that even in cases with initially deemed unresectable, long-term survival may be achievable through conversion surgery following chemotherapy.

## 1. Introduction

Biliary tract cancer comprises cancers with different backgrounds that arise from the biliary tract, including the biliary tree, gallbladder, and papilla of Vater.

The incidence of cholangiocarcinoma is increasing, and most biliary tract cancers are still found in advanced stages, despite improvements in endoscopic diagnostic ability [1,2,3]. The prognosis of biliary tract cancer remains poor, with a 5-year survival rate of 5–24% for cholangiocarcinoma, 24–39% for gallbladder cancer, and 45–61% for ampullary carcinoma [4,5,6,7], but gradual improvements are being made [4,8,9].

Gemcitabine- or 5-fluorouracil-based chemotherapy are the recommended treatment regimens [10]. To date, the treatment effects of chemotherapy, molecularly targeted therapy, and immunotherapy for unresectable biliary tract cancer have been limited, with a median survival time of 5–20 months [11,12,13,14]. Thus, long-term survival is not generally expected after these treatments, with only a few exceptions, showing a dramatic response to treatment [2,11,15]. Therefore, we administered down-staging chemotherapy for locally advanced or metastatic biliary tract cancer and initialized conversion surgery for patients in whom R0 resection became feasible and who were able to tolerate surgical resection from 2007. Several studies have demonstrated the efficacy of conversion surgery for pancreatic cancer, and long-term survival has been achieved in some cases [16,17,18]. We have accumulated cases with conversion surgery in response to these previous reports. This study aimed to clarify the clinical significance of conversion surgery in patients with initially unresectable biliary tract cancer.

## 2. Materials and Methods

### 2.1. Study Population

The focus of the present study was biliary tract cancer rather than bile duct cancer. Since ampullary carcinoma is classified as a type of biliary tract cancer in the Japanese classification of biliary tract cancer, it was included in this study [4]. The participants in the present study were Japanese patients with pathologically confirmed biliary tract cancer—including intrahepatic cholangiocarcinoma, perihilar cholangiocarcinoma, distal bile duct cancer, gallbladder cancer, and ampullary carcinoma—who were treated with surgery or chemotherapy between July 2007 and January 2018. Disease classification and staging of these biliary tract cancers were performed according to the Union for International Cancer Control, 8th edition [19]. The reason for the unresectability of biliary tract cancer was classified as locally advanced or metastatic disease. Locally advanced disease was defined as biliary tract cancer that met the following criteria: arterial invasion in which the artery could not be reconstructed, portal vein invasion in which the vein could not be reconstructed, invasion to the roots of all three hepatic veins, broad tumor involvement in bilateral intrahepatic bile ducts, tumor extension more distal than 10 mm from both the U point (the inflection point between the umbilical portion and transverse sections in the left portal vein) and the *p* point (the junction between the anterior and posterior branches of the right portal vein), or insufficient hepatic reserve to allow the patient to endure hepatic resection [20,21,22,23,24]. The hepatic reserve was assessed using ICGK and the plasma disappearance rate of indocyanine green [25]. When the remnant ICGK, expressed as ICGK × % remnant liver volume (ICGKF), was less than 0.05, which is known as the Nagoya criteria, the case was considered to have an insufficient hepatic reserve [26]. The ICGKF calculation for patients requiring portal vein embolization was performed 3 weeks after the embolization. All decisions regarding the initially unresectable biliary tract cancer were made at Yokohama City University. The regional lymph nodes were defined according to the primary site of the tumor. For intrahepatic and perihilar cholangiocarcinoma, the regional lymph nodes included the hilar (common bile duct, hepatic arteries, portal vein, and cystic duct), periduodenal, peripancreatic, and gastrohepatic lymph nodes. For distal bile duct cancer and gallbladder cancer, the regional lymph nodes included the hilar, celiac trunk, peripancreatic, gastrohepatic lymph nodes and nodes along the superior mesenteric artery. For ampullary carcinoma, the regional lymph nodes included nodes along with common bile duct, common hepatic duct, portal vein, celiac trunk, superior mesenteric artery, and peripancreatic and gastrohepatic lymph nodes. Judgements of locally advanced disease were made using diagnostic imaging such as direct cholangiography, abdominal computed tomography (CT) including 3-dimensional imaging and multi-planar reconstruction, magnetic resonance imaging (MRI), or positron emission tomography (PET)-CT. Distant metastases, such as distant lymph node metastasis beyond the regional lymph nodes, liver metastasis, lung metastasis, bone metastasis, and peritoneal dissemination, were made by CT, MRI, PET-CT, or surgical exploration during laparotomy. For example, para-aortic lymph node metastasis was diagnosed preoperatively when PET-CT showed abnormal uptake in the para-aortic lymph nodes or when lymph nodes from the hepatoduodenal ligament to the para-aortic region were continuously enlarged. Intraoperative diagnosis was histologically confirmed by para-aortic lymph node sampling. Peritoneal dissemination and liver metastases were also diagnosed using preoperative imaging or intraoperative rapid histological examination. If these distant metastases were detected intraoperatively, surgical resection was halted, and down-staging chemotherapy was initiated as soon as possible.

As neoadjuvant chemotherapy (NAC), three courses of gemcitabine + S-1 therapy were administered to patients with biliary tract cancer with regional lymph node metastasis and tumor involvement of the hepatic artery and/or portal vein or Bismuth type 4 on preoperative imaging [27]. As for down-staging chemotherapy, gemcitabine-based chemotherapy was administered to the patients with unresectable biliary tract cancer. The efficacy of chemotherapy was determined by tumor markers and diagnostic imaging modalities such as CT, MRI, or PET-CT after every 2–3 chemotherapy courses and classified according to the Response Evaluation Criteria in Solid Tumors (RECIST) criteria [28]. Conversion surgery was indicated for initially unresectable biliary tract cancer patients for whom the cancer did not develop progressive disease (PD) after chemotherapy, namely, those who responded to chemotherapy by achieving stable disease (SD) or partial response (PR) according to the RECIST criteria and were considered candidates for R0 resection. These patients finally underwent conversion surgery after careful consideration of the potential benefits and surgical risks, including tolerance to surgery, and provision of adequate informed consent. Staging for patients with initially unresectable biliary tract cancer who underwent conversion surgery was performed at three different time points: (1) at the time of first diagnosis of the cancer as unresectable based on preoperative imaging or laparotomy findings and biopsy, (2) after chemotherapy before conversion surgery on preoperative imaging, and (3) after conversion surgery based on the pathological diagnosis of the resected specimen. Adjuvant chemotherapy was administered to patients with a stable general condition. This study was conducted in accordance with the Declaration of Helsinki and approved by the Institutional Review Board of Yokohama City University (IRB number: B190300039).

### 2.2. Statistical Analysis

Statistical analyses were conducted using SPSS Statistics version 21.0 (IBM, New York City, NY, USA). Continuous variables are presented as means and ranges. Survival time was calculated from the initial day of chemotherapy treatment. Survival analyses were performed using the Kaplan–Meier method and compared using the log-rank test. Statistical significance was set at *p* < 0.05.

## 3. Results

### 3.1. Patients

During the study period, 527 patients were pathologically diagnosed with biliary tract cancer and were referred to our department for surgical treatment (Figure 1).

In this cohort, 340 patients were scheduled for straightforward resection, of whom 317 underwent radical resection. The remaining 23 patients were diagnosed intraoperatively as unresectable, and systemic chemotherapy was administered to all 23 patients. The reasons for the diagnosis of unresectable biliary tract cancer were liver metastasis (n = 5), peritoneal dissemination (n = 8), distant lymph node metastasis (n = 7), and vascular invasion where the vascular reconstruction was not feasible (n = 4). One of these patients developed metastases in multiple organs. Five out of the 23 cases of unresectable biliary tract cancer were converted to surgery after chemotherapy. NAC was administered to 139 of the 527 patients with biliary tract cancer, and 109 patients underwent radical resection after NAC. The remaining 30 patients were diagnosed as unresectable during NAC or at the time of surgery. The reasons for the diagnosis of unresectable biliary tract cancer were liver metastasis (n = 11), peritoneal dissemination (n = 9), distant lymph node metastasis (n = 5), lung metastasis (n = 1), and extensive tumor involvement in the bile ducts (n = 7). Two of these patients developed metastasis in multiple organs. Fifteen cases were determined to be unresectable due to disease progression during NAC, while the remaining 15 cases were found to have liver metastases, peritoneal dissemination, or para-aortic lymph node metastases at laparotomy, leading to a diagnosis of unresectable biliary tract cancer. A total of 3 of these 30 patients with unresectable biliary tract cancer underwent conversion surgery. The remaining 48 of the 527 biliary tract cancer cases were diagnosed as unresectable at the initial visit, and 12 patients underwent conversion surgery following chemotherapy. As a result, 20 of the 101 initially unresectable biliary tract cancers finally underwent conversion surgery, with an overall conversion rate of 19.8%.

### 3.2. Characteristics of Patients with Unresectable Biliary Tract Cancer (n = 101)

The patient characteristics are summarized in Table 1.

The mean age was 66.5 years, and the patients included 69 men and 32 women. The anatomical tumor locations were as follows: intrahepatic cholangiocarcinoma in 35 cases, perihilar cholangiocarcinoma in 40 cases, distal bile duct cancer in 2 cases, gallbladder cancer in 20 cases, and ampullary carcinoma in 4 cases. A total of 27 cases were unresectable due to locally advanced disease, whereas 74 were unresectable due to metastatic disease. The sites of distant metastasis were the liver in 30 cases, distant lymph nodes in 23 cases, peritoneal dissemination in 23 cases, the lungs in 7 cases, and bone in 4 cases. Eleven patients had multiple distant metastases. The most common chemotherapy regimen is gemcitabine plus cisplatin (CDDP), followed by gemcitabine plus S-1. Forty-eight patients were treated with more than two kinds of regimen. Concomitant radiation therapy was administered to six patients. The best overall response evaluated with the RECIST criteria resulted in a complete response in 1 case, PR in 9 cases, SD in 45 cases, and PD in 45 cases. The response rate was 10%. Treatment response in four cases could not be evaluated due to disease progression or deterioration of the general condition. The median observation period for the 101 cases of unresectable biliary tract cancer was 11.1 months.

### 3.3. Details of Patients with Initially Unresectable Biliary Tract Cancer Who Underwent Conversion Surgery (n = 20)

Table 2 shows the details of the 20 unresectable biliary tract cancer cases in which the patient underwent conversion surgery.

The patients in cases 9, 12, and 14 received down-staging chemotherapy after NAC and ultimately underwent conversion surgery. Six cases showed locally advanced disease, including two cases with invasion to an artery, one case with invasion to a portal vein, four cases with invasion to the inferior vena cava, and one case with invasion to the hepatic vein. No patient with extensive biliary tract invasion or insufficient remnant liver volume was treated using conversion surgery. Among the remaining 14 cases of unresectable biliary tract cancer due to distant metastasis, liver metastasis was observed in 6 cases, para-aortic lymph node metastasis in 5 cases, and peritoneal dissemination in 3 cases. The chemotherapy regimens were as follows: gemcitabine plus CDDP in 14 patients, gemcitabine plus S-1 in 6 patients, and gemcitabine monotherapy in 1 patient. The mean number of chemotherapy cycles was 9.8, and the mean duration of chemotherapy was 7.8 months. Treatment response according to the RECIST criteria resulted in PR in 7 cases and SD in 13 cases. Diagnostic imaging confirmed down-staging after chemotherapy in 13 patients. The conversion rates according to reasons for unresectability were as follows: 22% with locally advanced disease, 20% with liver metastasis, 22% with distant lymph node metastasis, and 13% with peritoneal dissemination. Patients with lung, bone, or multiple distant organ metastases did not undergo conversion surgery.

As for surgical procedure, combined resection and reconstruction of an artery, portal vein, inferior vena cava, or hepatic vein was performed in four, eight, and five cases, respectively. Combined resection of other organs was required in three cases. The mean operative time was 823 min, and the mean intraoperative blood loss was 1902 mL. In terms of short-term outcomes, postoperative complications of grade IIIa or higher according to the Clavien–Dindo classification occurred in 10 cases, with no mortality within 90 days after radical resection. The mean postoperative hospital stay was 31 days.

Pathological R0 resection was achieved in 17 cases, while microscopic residual tumor (R1) was observed in 3 cases. Moreover, the final diagnosis based on pathological examination confirmed that 15 cases were down-staged compared with the pre-treatment condition. Case 3, with locally advanced disease, was finally diagnosed as stage IV because of pathologically confirmed distant metastasis to the gallbladder. Lesions of liver metastasis disappeared in three of the six cases with liver metastasis. In case 12, a 2 mm metastatic lesion was found in the liver during the first laparotomy. The lesion was resected for intraoperative frozen biopsy, and no distant metastases were identified during radical resection after chemotherapy. A few localized peritoneal disseminations were found during the first laparotomy in all three patients with peritoneal dissemination, and these disseminated lesions were resected for intraoperative frozen biopsy. After chemotherapy, no new peritoneal metastases were found in two cases, whereas localized dissemination was still found and resected again in one case after radical resection. In three of the five cases with para-aortic lymph node metastasis, disappearance of para-aortic lymph node metastasis was observed on imaging after chemotherapy. The results for pathological para-aortic lymph node metastasis were negative on lymph node sampling at the time of radical resection. In the remaining two cases (cases 18 and 19), para-aortic lymph node metastasis was diagnosed by lymph node sampling during the first surgery, and no para-aortic lymph node metastasis was found at the time of radical resection after chemotherapy. In case 20, multiple lymph node swellings including regional and para-aortic lymph nodes were detected and diagnosed as perihilar cholangiocarcinoma, T2bN2M1. After down-staging chemotherapy, the swelled regional lymph nodes remained, while the para-aortic lymph node swelling disappeared on imaging. Finally, these regional lymph nodes resected during surgery showed no histological evidence of metastasis, indicating that chemotherapy was highly effective against lymph node metastasis. However, axial lymph node recurrence occurred 11 months after conversion surgery.

Adjuvant chemotherapy, either gemcitabine-based or S-1, was administered to 17 out of 20 patients who underwent conversion surgery. Recurrence was observed in 14 patients. The survival times ranged from 10.5 to 113.5 months.

### 3.4. Survival Comparison Between Patients with Unresectable Biliary Tract Cancer Who Underwent Conversion Surgery and Those Treated by Chemotherapy Alone

Survival curves from the time of chemotherapy induction comparing 20 cases in the conversion surgery group and 81 cases in the chemotherapy-only group are shown in Figure 2.

Survival was significantly longer in patients treated with conversion surgery than in those who did not have conversion surgery (*p* < 0.001). The 3- and 5-year survival rates in patients who underwent conversion surgery were 65.0% and 55.0%, respectively, compared to 10.3% and 8.6%, respectively, in patients who did not undergo conversion surgery.

The 3- and 5-year survival rates and median survival time were significantly better in the conversion surgery group (65.0%, 55.0%, and 98.8 months [range: 10.5–113.5 months], respectively) than in the chemotherapy-only group (10.3%, 8.6%, and 11.1 months [range: 0.6–89.0 months], respectively; *p* < 0.001). Next, a subgroup analysis based on the reason for unresectability was performed (Figure 3).

Conversion surgery was associated with significantly longer overall survival in patients with locally advanced disease, liver metastasis, distant lymph node metastasis, and peritoneal dissemination.

## 4. Discussion

Gemcitabine-based chemotherapy, particularly gemcitabine plus cisplatin (CDDP), is a standard treatment for unresectable biliary tract cancer, with a median survival time of 8 months [2,11,29]. Recently, combination therapy with gemcitabine, CDDP, and S-1 has shown additional effects over gemcitabine plus CDDP therapy, with a high response rate of 41.5%, which is considerably higher than the response rate of 10% seen in the present study [30]. However, randomized controlled trials using combination treatment with molecularly targeted therapies such as erlotinib, panitumumab, or sorafenib to gemcitabine-based chemotherapy have failed to show an additional benefit [11]. Several reports have demonstrated that actionable mutations are present in 25–50% of biliary tract cancers [31,32]. Positive results for *fibroblast growth factor 2* and *isocitrate dehydrogenase 1/2* mutations have been reported in 11–45% and 10–20% of intrahepatic cholangiocarcinoma [33]. Some molecularly targeted drugs for biliary tract cancers with actionable mutations have shown clinical efficacy, particularly *FGFR*-targeted therapy for *FGFR*-mutant intrahepatic cholangiocarcinoma, achieving an objective response rate of 20–50% [31]. DNA mismatch repair protein-deficient and/or microsatellite instability-high tumors represent less than 2% of biliary tract cancers. A subset analysis of the KEYNOTE-158 study revealed that treatment with pembrolizumab, a programmed cell death-1 inhibitor, for previously treated advanced biliary tract cancer with a deficiency in DNA mismatch repair/high microsatellite instability resulted in favorable outcomes, with a high response rate of 40% and a median survival time of 20 months [14,31]. Considering these promising results, active genomic analyses are essential for selecting the best treatment option for biliary tract cancers. As the number of patients with biliary tract cancer responding to chemotherapy and/or molecularly targeted therapy increases, more patients are expected to be treated using conversion surgery.

The efficacy of conversion surgery for biliary tract cancer has recently been reported in several studies. Kato et al. first reported the results of conversion surgery for locally advanced biliary tract cancer after gemcitabine treatment [34]. Some 8 of the 22 (36%) patients with locally advanced biliary tract cancer underwent conversion surgery, including combined resection of the inferior vena cava in 3 cases. R0 resection was performed in four patients, and the median survival time in patients treated with conversion surgery was 19 months, comparable to that of resectable biliary tract cancer cases treated using radical surgery. A subsequent report compared the outcomes of conversion surgery for locally advanced biliary tract cancer after gemcitabine or gemcitabine plus CDDP treatment [35]. No significant difference in overall survival was observed between chemotherapy regimens, but gemcitabine plus CDDP treatment was associated with a stronger pathological response. In a multi-institutional retrospective study, Noji et al. administered gemcitabine-based chemotherapy to patients with 110 unresectable biliary tract cancers, and 24 cases—comprising 8 cases with locally advanced disease, 4 cases with bulky lymph node metastasis, and 12 cases with distant metastasis—underwent conversion surgery and achieved an R0 resection rate of 83% [36]. They reported favorable survival in patients who underwent conversion surgery, with a 5-year overall survival rate and median survival time of 42% and 34 months, respectively. Moreover, the subgroup analysis revealed that conversion surgery was associated with longer survival in patients with locally advanced disease, bulky lymph node metastasis, and distant metastatic disease.

Moreover, subgroup analysis in the present study demonstrated the efficacy of conversion surgery for patients with liver metastasis, distant lymph node metastasis, peritoneal dissemination, and locally advanced disease. Regarding liver metastasis, metastatic lesions were not present in 4 of the 6 cases at radical resection. Liver metastases were removed by surgical resection in the remaining two cases, one of which had two liver metastases and the other showed liver metastasis. Morino et al. reported that locoregional treatment for biliary tract cancer patients with single-organ recurrence within three metastatic lesions was associated with longer survival, suggesting that conversion surgery may be beneficial for biliary tract cancer patients showing oligometastases [37]. These results indicate that conversion surgery for biliary tract cancer with oligometastases is effective and can be interpreted as showing some chance of a cure. Recently, the ASCOT trial revealed that adjuvant S-1 therapy significantly improved survival in Asian patients with resected biliary tract cancer [38]. Adjuvant chemotherapy may contribute to favorable long-term outcomes in the patients who underwent conversion surgery in the present study.

Careful attention should be paid to the safety of conversion surgery for initially unresectable biliary tract cancer, particularly in locally advanced disease where more invasive surgical procedures, such as combined resection of the inferior vena cava, are required. Furthermore, consideration of chemotherapy-induced hepatotoxicity is essential in conversion surgery for biliary tract cancer, where liver damage may already be present due to preoperative jaundice or cholangitis [39]. Matsuyama et al. demonstrated the efficacy of NAC using gemcitabine plus S-1 for perihilar cholangiocarcinoma and reported that preoperative cholangitis occurred in 45% of the patients [27]. Since stabilization of the general condition of patients by controlling cholangitis is indispensable for performing conversion surgery, close collaboration with gastroenterologists and endoscopists is required to provide treatment as a multidisciplinary team. Kato et al. reported no mortality after conversion surgery for locally advanced biliary tract cancer [34,35]. Noji et al. reported that 67% of patients suffered from grade III or higher postoperative complications, and one patient died of postoperative liver failure after right hemihepatectomy [36]. No mortality was observed in the present study, but grade III or higher postoperative complications, including liver failure and septic shock, occurred in 50% of the patients. Therefore, the indications for conversion surgery for biliary tract cancer should be decided after careful assessment of the risks and benefits and adequate informed consent. Since the operative mortality rate after highly invasive hepato-biliary–pancreatic surgeries is relatively low in high-volume centers, conversion surgery should be performed in institutions where careful perioperative management can be provided [40,41,42].

One important limitation of the present study was that the exact therapeutic effect of conversion surgery itself could not be evaluated because only selected biliary tract cancer patients who maintained their performance status could tolerate surgery, and only those who responded well to chemotherapy underwent conversion surgery.

## 5. Conclusions

Conversion surgery for initially unresectable biliary tract cancer results in a high R0 resection rate and is associated with favorable overall survival, even among patients with oligometastases. Moreover, conversion surgery is comparatively safe despite its high surgical invasiveness. Our results support conversion surgery for initially unresectable biliary tract cancer, particularly for intrahepatic cholangiocarcinoma or perihilar cholangiocarcinoma, in selected patients.

## Figures and Tables

**Figure 1 cancers-17-00873-f001:**
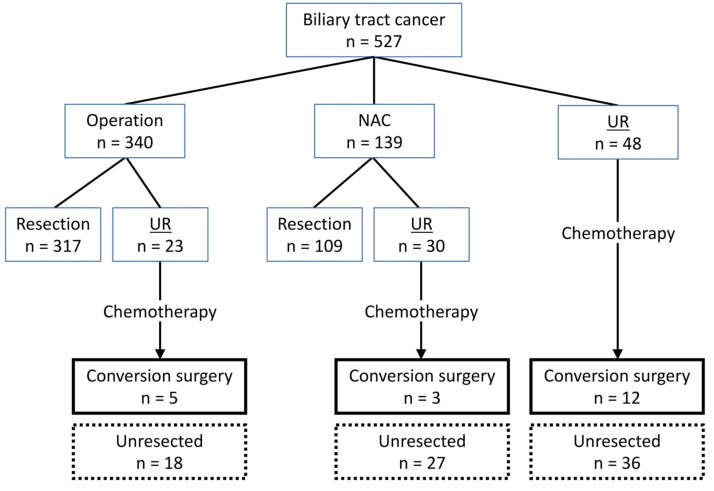
Flow chart of enrolled patient selection. NAC, neoadjuvant chemotherapy; UR, unresectable. The underlines were used to emphasize ‘UR’.

**Figure 2 cancers-17-00873-f002:**
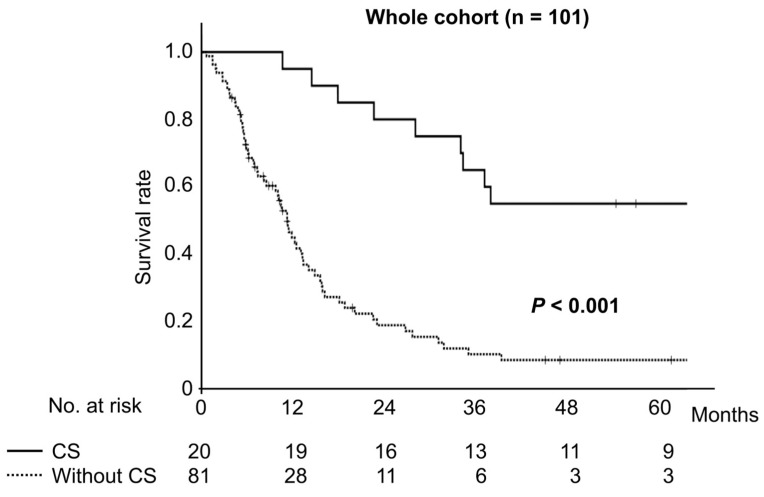
Comparison of overall survival from the initial treatment day between patients treated with conversion surgery and those who did not undergo conversion surgery.

**Figure 3 cancers-17-00873-f003:**
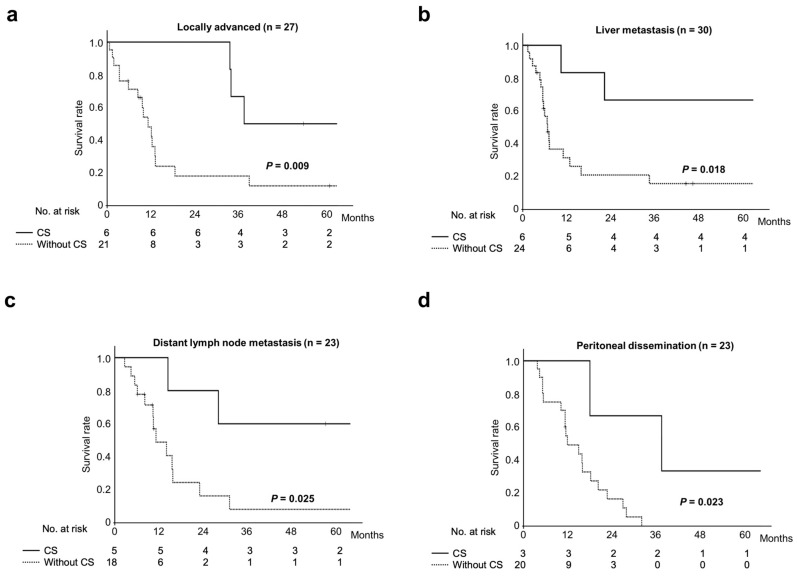
Comparison of overall survival between groups classified by reason for unresectability. Conversion surgery was associated with better overall survival in patients with (**a**) locally advanced disease (*p* = 0.009); (**b**) liver metastasis (*p* = 0.018); (**c**) distant lymph node metastasis (*p* = 0.025); and (**d**) peritoneal dissemination (*p* = 0.023).

**Table 1 cancers-17-00873-t001:** Characteristics of patients with initially unresectable BTC.

Factors	Initially Unresectable BTC (n = 101)
Age (years)	66.5 (34–84)
Sex	
Male	69
Female	32
Anatomical tumor location	
Intrahepatic cholangiocarcinoma	35
Perihilar cholangiocarcinoma	40
Distal bile duct cancer	2
Gallbladder cancer	20
Ampullary carcinoma	4
Reason for unresectability	
Locally advanced disease	27
Metastatic disease	74
Liver metastasis	30
Lung metastasis	7
Bone metastasis	4
Distant lymph node metastasis	23
Peritoneal dissemination	23
Treatment regimen	
Gem + CDDP	79
Gem + S-1	56
Gem	16
Radiation	
Yes	6
No	0
Cycles of chemotherapy	13.7 (1–89)
RECIST	
Complete response	1
Partial response	9
Stable disease	45
Progressive disease	42
Not evaluated	4
Conversion surgery	
Yes	20
No	81

BTC, biliary tract cancer; Gem, gemcitabine; CDDP, cisplatin; RECIST, Response Evaluation Criteria in Solid Tumors.

**Table 2 cancers-17-00873-t002:** Details of patients who underwent conversion surgery.

Case	Age, Sex	Tumor Location	Reason for Unresectability	Staging at Diagnosis	Regimen	RECIST	Staging After Chemotherapy	Surgical Procedure	Final Staging	Residual Tumor	Complications ≥ Grade IIIa	Recurrence	Survival
1	40s, female	IHCC	Locally advanced (A, PV)	T3N1M0, Stage IIIB	Gem + CDDP × 8	PR	T3N0M0, Stage IIIA	LT + C + BDR	T1aN0M0, Stage IA	R0	Abdominal abscess	No	53.9 months, alive
2	50s, female	IHCC	Locally advanced (IVC, HV)	T4N0M0, Stage IIIB	Gem + CDDP × 8	SD	T4N0M0, Stage IIIB	RT + C + BDR + PVR + IVCR + HVR	T3N0M0, Stage IIIA	R0	-	Yes	33.7 months, dead
3	50s, male	IHCC	Locally advanced (A)	T4N1M0, Stage IIIB	Gem + CDDP × 10	SD	T3N1M0, Stage IIIB	RH + C + BDR + PVR	T3N1M1, Stage IV	R1	Abdominal abscess	Yes	34.0 months, dead
4	30s, female	IHCC	Locally advanced (IVC)	T4N1M0, Stage IIIB	Gem × 11	SD	T4N0M0, Stage IIIB	LH + C + BDR + IVCR	T1aN0M0, Stage IA	R0	-	Yes	113.5 months, alive
5	50s, female	IHCC	Locally advanced (IVC)	T4N1M0, Stage IIIB	Gem + CDDP × 16	PR	T3N0M0, Stage IIIA	LH + C + BDR + AR	T2N0M0, Stage II	R0	Bowel obstruction	Yes	84.3 months, alive
6	50s, female	IHCC	Locally advanced (IVC)	T4N0M0, Stage IIIB	Gem + CDDP × 24	PR	T4N0M0, Stage IIIB	RH + C + BDR + IVCR + Sp	T2N1M0, Stage IIIB	R1	Liver failure, Bile leakage	No	37.6 months, dead
7	70s, female	Perihilar CC	Liver metastasis	T3N1M1, Stage IVB	Gem + S-1 × 3	SD	T2bN1M1, Stage IVB	RH + C + BDR + PVR	T2bN1M1, Stage IVB	R0	-	No	94.2 months, alive
8	60s, male	Distal BDC	Liver metastasis	T3N1M1, Stage IV	Gem + S-1 × 9	PR	T3N1M0, Stage IIIC	PD	T2N1M0, Stage IIB	R0	POPF	Yes	22.4 months, dead
9	60s, female	GBC	Liver metastasis	T4N1M1, Stage IVB	Gem + S-1 × 3 Gem + CDDP × 3	SD	T4N1M1, Stage IVB	Sec + C + BDR + Du + Co	T4N1M1, Stage IVB	R0	Abdominal abscess	Yes	10.5 months, dead
10	70s, female	GBC	Liver metastasis	T4N1M1, Stage IVB	Gem + S-1 × 6	PR	T3N1M0, Stage IIIB	Seg + BDR	T2bN0M0, Stage IIB	R0	-	No	92.5 months, alive
11	70s, female	Ampullary Ca	Liver metastasis	T2N0M1, Stage IV	Gem + S-1 × 4 S-1 × 5	PR	T2N0M0, Stage IB	PD	T1aN0M0, Stage IA	R0	POPF	No	87.5 months, alive
12	80s, male	Perihilar CC	Liver metastasis	T3N1M1, Stage IVB	Gem + CDDP × 6	SD	T3N0M0, Stage IIIA	EHBDR + IVCR + Co	T2aN0M0, Stage II	R0	-	Yes	98.8 months, dead
13	70s, female	Perihilar CC	Peritoneal dissemination	T4N1M1, Stage IVB	Gem + CDDP × 8	SD	T4N1M0, Stage IIIC	LH + C + BDR + AR + PVR	T2bN0M0, Stage II	R0	Abdominal abscess	No	66.2 months, alive
14	60s, female	Perihilar CC	Peritoneal dissemination	T3N2M1, Stage IVB	Gem + CDDP × 8	SD	T3N1M0, Stage IIIC	RH + C + BDR + PD + PVR	T2bN1M0, Stage IIIC	R0	-	Yes	17.7 months, dead
15	50s, female	IHCC	Peritoneal dissemination	T2N1M1, Stage IV	Gem + CDDP × 8	PR	T2N1M0, Stage IIIB	LH + C + BDR + AR	T1aN0M1, Stage IV	R0	-	Yes	36.8 months, dead
16	60s, female	IHCC	Para-aortic lymph node metastasis	T4N1M1, Stage IV	Gem + S-1 × 6	SD	T4N1M0, Stage IIIB	RT + C + BDR + PVR + IVCR	T4N1M0, Stage IIIB	R0	-	Yes	91.7 months, alive
17	50s, male	Perihilar CC	Para-aortic lymph node metastasis	T3N2M1, Stage IVB	Gem + CDDP × 23	SD	T3N2M1, Stage IVB	RT + C + BDR + AR + PVR	T2bN2M0, Stage IVA	R0	Enteritis	Yes	67.4 months, alive
18	70s, male	Perihilar CC	Para-aortic lymph node metastasis	T3N2M1, Stage IVB	Gem + CDDP × 9	SD	T3N2M0, Stage IVA	RH + C + BDR + PVR	T3N2M0, Stage IVA	R1	Postoperative bleeding	Yes	27.8 months, alive
19	70s, male	Perihilar CC	Para-aortic lymph node metastasis	T2aN2M1, Stage IVB	Gem + CDDP × 8	SD	T2aN2M0, Stage IVA	EHBDR + PD	T2aN2M0, Stage IVA	R0	Cholangitis, septic shock	Yes	14.3 months, dead
20	60s, male	Perihilar CC	Para-aortic lymph node metastasis	T2bN2M1, Stage IVB	Gem + CDDP × 10	SD	T2bN2M0, Stage IVA	Sec + C + BDR + PD	T1N0M0, Stage I	R0	-	Yes	56.5 months, alive

IHCC, intrahepatic cholangiocarcinoma; CC, cholangiocarcinoma; BDC; bile duct cancer; GBC gallbladder cancer; A; artery; PV, portal vein; PR, partial response; SD, stable disease; LT, left hepatic trisectionectomy; C, caudate lobectomy; BDR, bile duct resection; RT, right hepatic trisectionectomy; PVR, portal vein resection; IVCR, resection of the inferior vena cava; HVR, resection of the hepatic vein; RH, right hemihepatectomy; LH, left hemihepatectomy; AR, artery resection; Sp, splenectomy; PD, pancreaticoduodenectomy; Sec, hepatic sectionectomy; Du, partial resection of the duodenum; Co, partial colectomy; Seg, hepatic segmentectomy; EHBDR; extended hilar bile duct resection.

## Data Availability

The datasets generated and/or analyzed during the current study are available from the corresponding author on reasonable request.

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
