# Peer review of "Efficacy of Conversion Surgery for Initially Unresectable Biliary Tract Cancer That Has Responded to Down-Staging Chemotherapy"

_cancers, 2025, doi:10.3390/cancers17050873_

Round 1

Reviewer 1 Report

Comments and Suggestions for Authors

Introduction

Chronological Inconsistency
The authors' statement about being inspired by conversion surgery results for pancreatic cancer (Page 2, Lines 56-58) is chronologically inconsistent with their study period (2007-2018). The first report on CS for pancreatic cancer was published in 2011 by Kato et al. This section should be revised to accurately reflect the timeline and inspiration for their study.Chemotherapy Regimen
The authors' assertion that 5-FU regimens are common for biliary malignancies (Page 2, Line 50) is not accurate on a global scale. This statement should be revised to reflect more widely accepted treatment protocols.

Materials and Methods

Tumor Involvement Definition
The concept of "broad tumor involvement bilateral" (Page 2, Line 74) requires clarification. To improve this, the authors should:

  1. Reference Noji et al.'s "limit of dividing" for each surgical procedure (Noji et al., JGIS 2021).
  2. Cite previous reports from their institute (Shimada et al., extended bile duct resection) and similar studies (Noji et al., JGIS 2014; Onoue et al.) to illustrate the limits of bile duct resection without hepatectomy.

ICGK Explanation
An explanation for ICGK (Page 2, Line 79) should be provided for clarity.ICGKF Criteria
The phrase "ICGK and percentage of future remnant liver volume less than 0.05" (Page 2, Line 80) is ambiguous. It appears the authors are referring to ICGK × % Remnant Liver Volume (RLV): ICGKF < 0.05, known as the Nagoya criteria (advocated by Prof. Nagino and Prof. Yokoyama). This should be clearly stated.Portal Vein Embolization
The timing of ICGKF calculation for patients requiring portal vein embolization should be specified.Regional Lymph Node Definition
The authors should clarify that the definition of regional lymph nodes (Page 2, Line 87) varied based on tumor location.

Results

Figure 1 Clarifications
Figure 1 requires several clarifications:

  1. The indication for neoadjuvant chemotherapy (NAC) is unclear.
  2. The classification of 30 out of 139 patients as unresectable (UR) needs explanation.
  3. Clarify whether UR patients after NAC were initially unresectable or if the timing of transitioning from chemotherapy to surgery was misjudged.
  4. Explain how three UR patients after NAC could undergo surgery, including indications, regimen, and duration of NAC.
  5. For upfront surgery (n=340), provide details on why 23 of 340 patients were diagnosed as UR.

Survival Time Reporting
The sentence on Page 7, Line 217 should be revised to accurately report median survival time, e.g., "median survival time was XX months (range: @@-@@)".

Discussion

The first paragraph regarding chemotherapy should be removed as it does not contribute significantly to the discussion of the study's findings.

Reviewer 2 Report

Comments and Suggestions for Authors

conversion surgery of biliary cancer. It isn't easy to analyze, but thank you for organizing good results on an important topic. There will be a lot of research in the future, and I think your report will be a valuable basis. I will ask you some (minor) questions to help readers understand, including me.

1. Is it right to include ampullary cancer to your research because ampullary cancer has many different characteristics from bile duct cancer? (The title and main issue of the discussion are also deal with bile duct cancer.)

2. decision of 'initial unsustainable' due to locally advanced tumor --> all done only at the author's hospital? This is because the possibility of surgery can be viewed differently depending on experience or policy from the hospital or doctor when making a diagnosis.)

3. paraaortic LN metastasis in staging: diagnosed by radiological? Or is tissue confirmed?
(Because there are sometimes false positives in imaging findings)
And when trying to curative resection, do you stop when paraortic LN sampling was positive?

4. case20 : N2M1 (even though N2 after CTx) - final N0

    after reviewing of this case through author's study, was there over-estimation for nodal assessment?

5. "Staging was performed at three different times" -- Does it mean surgery itself or diagnostic evaluation?
6. What do authors think about the possibility that adjuvant CTx affected the good 5yr-survival results?
